# The Effectiveness of Mental Health First Aid Training among Undergraduate Students in Switzerland: A Randomized Control Trial

**DOI:** 10.3390/ijerph20021303

**Published:** 2023-01-11

**Authors:** Shota Dzemaili, Jérôme Pasquier, Annie Oulevey Bachmann, Meichun Mohler-Kuo

**Affiliations:** 1La Source, School of Nursing Sciences, HES-SO University of Applied Sciences and Arts of Western Switzerland, 1004 Lausanne, Switzerland; 2Center for Primary Care and Public Health (Unisanté), University of Lausanne, 1011 Lausanne, Switzerland; 3Department of Child and Adolescent Psychiatry and Psychotherapy, University Hospital of Psychiatry Zurich, University of Zurich, 8032 Zurich, Switzerland

**Keywords:** mental health first aid, undergraduate students, effectiveness, mental health, knowledge, attitude, behavior, RCT

## Abstract

Half to three-fourths of mental disorders appear during adolescence or young adulthood, and the treatment gap is mainly due to lack of knowledge, lack of perceived need, and the stigmatization of mental illness. The aims of this study were to implement and evaluate a Mental Health First Aid (MHFA) training program among undergraduates. Participants were second-year students from two universities in the French-speaking region of Switzerland (*N* = 107), who were randomly assigned to an intervention group (*n* = 53) or control group (*n* = 54). The intervention group received a 12-h MHFA course. Online questionnaires were completed before the intervention (T0), and both 3 months (T1) and 12 months (T2) after the intervention in order to evaluate the participants’ mental health knowledge, recognition of schizophrenia, and attitudes and behaviors towards mental illness. We used Generalized Estimating Equations (GEE) to examine the effects of intervention over time. After the MHFA course, the intervention group showed significantly increased basic knowledge and confidence helping others with mental illness and reduced stigmatization at both T1 and T2 compared to their baseline scores and compared to control groups. This suggests that the MHFA training program is effective and has significant short-term and long-term impacts, in terms of enhancing basic knowledge about mental health and improving attitudes towards mental illness among undergraduate students.

## 1. Introduction

Epidemiological studies have revealed several important features of mental illness around the world. First, mental disorders, which span from youth through old age and include depression, anxiety, substance abuse, attention-deficit/hyperactivity disorder (ADHD) and dementia, are top contributors to the burden of disease worldwide [1,2,3,4]. In addition, mental illnesses are among the leading causes of disability-adjusted life years (DALYs) [2], the global burden of Common Mental Health Disorders (CMDs) continues to grow over time [3], and by 2030, depression is predicted to become the leading cause of disability in high-income countries [4]. Such disorders pose ever-increasing challenges to healthcare systems in both developed and developing regions. Second, there is an enduring worldwide treatment gap that leaves half of those affected with severe mental disorders without appropriate treatment [5,6,7,8], often creating several-year latencies between the onset of a disorder and a person’s first treatment contact [9,10,11]. Third, about half to three-fourths of mental disorders start during adolescence and young adulthood [12,13]. This age-of-onset distribution has been linked to critical phases of the human brain’s neurobiological maturation [14]. Furthermore, a negative correlation exists between the age at which someone is diagnosed with a mental illness and their likelihood of current treatment. In other words, young people who develop a mental disorder have a lower probability of receiving treatment and longer delays between the onset of their disease and their first treatment [9,10,11]. One systematic review highlights the high prevalence of depression among undergraduate university students. Adolescence and young adulthood must, therefore, be considered critical times for mental disease, and should thereby warrant special attention [15].

As elsewhere, the marked influence of mental disorders on societal health has also been documented in Switzerland. Evidence of this is that, among Swiss residents receiving disability compensation (IV-Rente), the proportion for whom mental illness was the stated cause of disability has increased dramatically, from 24.0% in 1995 to 48.0% in 2019 [16]. Moreover, mental illness exhibited the greatest annual growth among all reasons for disability compensation between 2000 and 2019 [16,17]. This increase was especially pronounced among young adults aged 18 to 29. Consistent with this increase, mental illness has become the single most prevalent reason for someone to receive disability compensation (IV-Rente) in Switzerland.

Evidence of a treatment gap has also been identified in Switzerland, with an estimated 15 years between the age at which mental illness was initially experienced (median = 20–25 years old) and the age when the patient first registered for disability benefits (median = 40–45 years old) among those who received disability benefits due to mental illness from 1992 to 2006 [18]. This indicates that the treatment of mental illness is not yet very effective. Moreover, the Swiss Youth Epidemiological Study on Mental Health, conducted in 2018 revealed that the most common barriers to seeking care in young Swiss adults reflected the same barriers found in other studies, namely a lack of perceived need [19] and the preference to manage the problem oneself [19,20]. Another reason for lack of seeking help includes mental illness stigma and lack of knowledge regarding mental health problems [21]. Therefore, an intervention that effectively improves mental health by increasing knowledge and awareness about symptoms that require professional mental health treatment is urgently needed.

Mental Health First Aid (MHFA) is an educational training program that teaches individuals how to recognize the signs of a mental illness (including substance use problems) and provide immediate help and support. It was originally developed in Australia by Prof. Anthony Jorm and Betty Kitchener in the year 2000. They recognized the need for early intervention, and that the general public could provide the first early intervention prior to contact with health professionals. This “first aid” is given until appropriate professional treatment is received or until the crisis resolves [22]. Already, MHFA has been proven effective at increasing knowledge regarding mental health, increasing someone’s confidence and intention to help, and reducing the stigma towards mental illness [23,24,25]. Notable impacts of MHFA include raising the level of awareness of the symptoms of mental disorders and potentially increasing the likelihood that students with mental health problems will ask for professional help and have better outcomes [26].

The MHFA program has been licensed and adapted in more than 25 countries worldwide. In June 2018, the MHFA was licensed to Switzerland via the Swiss association Pro Mente Sana, under the name “ensa”. To date, no study in Switzerland has examined the effectiveness of ensa, and few studies world-wide have examined its long-term effects or how they might be influenced by cultural differences [27,28,29,30,31,32]. Furthermore, attending university is a big change for many young adults [33,34]. They face many challenges due to academic pressure, changes in living arrangements, and new social relationships. Many college students experience mental and emotional difficulties [33,34,35]. Therefore, the aims of this study were (a) to implement the MHFA/ensa intervention for undergraduate university students, and (b) to examine its short-term and long-term effects in Switzerland by evaluating the effects of an MHFA/ensa training program on knowledge, attitude, behavior, and mental health status among undergraduate students at two Swiss universities of applied science.

## 2. Materials and Methods

### 2.1. Design

This interventional study was designed as a randomized controlled trial (RCT) in which participants were randomly assigned either to an intervention group that would receive the MHFA training course or to a control group without any intervention during the study period. However, the control group had the option to receive the MHFA course after the 12 months of study follow-up was completed. Both groups were assessed at three time points:

T0: Baseline survey from mid-September to October 2019, completed by both groups.

Intervention: MHFA/ensa training program from October 2019–December 2019.

T1: Post-intervention with first follow-up survey three months after the MHFA/ensa training intervention completed by both groups (from March to May 2020).

T2: Follow-up 12 months after the MHFA/ensa training intervention in December 2020 and completed by both groups.

### 2.2. Participants and Setting

Participants were full-time, second-year, undergraduate students who were at least 18 years old and attending either La Source School of Nursing Sciences (La Source) or The School of Management and Engineering, Vaud (HEIG-VD). Both schools are part of the University of Applied Sciences and Arts of Western Switzerland (Haute Ecole Spécialisée de Suisse Occidentale (HES-SO)).

### 2.3. Randomization

Participants were randomly (1:1) assigned to either the intervention group or the control group through computerized automated randomization once they returned the consent form and enrolled themselves for the online questionnaire. An independent market research institute, LINK (www.link.ch), was hired to set up an online platform and a computerized automated randomization procedure. This firm generated QR codes for students to access the online platform and randomized them into the two groups. Researchers were blinded to group allocation.

### 2.4. Recruitment and Procedure

Every eligible undergraduate student at La Source and HEIG-VD received information from their respective university about the study. In the meantime, posters, each universities’ website, and social media were used to promote the study.

A written invitation letter with a link and QR code was sent by mail to each student eligible to participate in the study. The mail also included detailed information about the study, a consent form, and an envelope for the latter to be returned. A second invitation letter was sent via email. Each participant received a username and a three-digit password to access the online survey. Those assigned to the intervention group were asked to choose class periods during which they could attend the program.

By October 2019, both the intervention and control groups completed a baseline survey developed to assess their basic knowledge about mental health and attitude towards mental illnesses, as well as their own mental health status. The MHFA/ensa training program was conducted among the intervention group from October to December 2019. Afterwards, the first follow-up survey was conducted 3 months after the intervention was completed, and a final follow-up survey was conducted 12 months after the intervention and roughly 13 months after the baseline survey for both groups. For each survey, both groups received emails with a link to respond the online questionnaires. Reminder emails were sent to those who had not yet responded to the assessment two weeks after the link was sent at each time point.

The recruitment procedure and participants are presented in Figure 1. Among 520 eligible students, 107 participants consented to participate and completed the baseline (T0) and were randomized to the intervention group (*n* = 53) or the control group (*n* = 54). The 3-month post-intervention survey (T1) was completed by 78 participants (intervention *n* = 36; control *n* = 42) with 27% of them failing follow up. The 12-month follow-up (T2) was completed by 71 participants (intervention *n* = 33; control *n* = 38). In general, students reacted very positively to our study. Non-participation was mainly due to conflict with their course schedule.

### 2.5. Ethics Consideration

The study was approved by the Cantonal (Canton of Vaud) Commission on Ethics in Human Research (CER-VD). Registration ID 2019-01296.

### 2.6. Intervention

The MHFA/ensa program is a 12-h face-to-face training program delivered in four sessions of three hours each across four consecutive weeks by two accredited MHFA/ensa instructors. The class was administered in small groups with a maximum of 10–15 students. The program included didactic lectures and “role playing” to teach students how to recognize and react to the signs and symptoms of mental health illness. Participants enrolled in the intervention group received an accompanying manual with content that covered helping people in mental health crises and/or the early stages of mental health problems. The mental health problems addressed during the training session included depressive symptoms, anxiety, psychotic disorders, and substance use disorders. The mental health crisis situations included suicidal thoughts/attempts and behaviors, acute stress reactions, panic attacks, and acute psychotic behaviors [22,36].

The intervention group attended the MHFA/ensa training program and answered an online questionnaire at three time-points (at baseline, 3 months after the intervention, and after 12 months).

Each session had two instructors who worked together to teach all four sessions over the four weeks. The intervention period took place from October 28 to the end of December 2019, depending on the students’ enrolment dates. Training sessions were offered at the end of the academic day, after students had completed their usual academic schedule.

Students’ academic schedule was not disrupted by MHFA/ensa training, which they attended after their usual class schedule voluntarily.

### 2.7. Outcome Measures

Knowledge was measured by assessing participants’ ability to identify a mental health problem and answer questions based on the content of the MHFA/ensa course. The questionnaire contained a short vignette presenting schizophrenia based on the Diagnostic and Statistical Manual of Mental Disorders (DSM-IV) (APA, 2000) and International Classification of Diseases (ICD-10) (WHO, 2016) [37] and validated by an Australian national sample of clinical psychologists, psychiatrists, and general practitioners [37,38,39]. For the schizophrenia vignette, inter-rater reliability with kappa for the correct responses was 0.96 [40]. It had already been used in various RCTs conducted worldwide [28,31,41]. In addition to this, eighteen statements assessed basic mental health knowledge based specifically on content taught in the ensa course and adapted from the National Survey of Mental Health Literacy [38].

Attitude was measured by assessing (a) stigmatizing attitudes based on the first part of the Depression Stigma Scale (DSS) on personal stigma, developed by Griffiths et al. [42], and the Social Distance Scale (SDS), developed by Link et al. [43]. This latter scale measures the social distance an individual wants to keep towards a person with a particular condition, such as a mental disorder [43]. These scales were adapted to be suitable for young adults [44]. The DSS personal stigma scale includes seven items, with responses provided on a 5-point Likert scale (higher scores imply higher stigma). Sufficient to good internal consistency and high test-retest reliability for the DSS has been reported [42,45,46]. The Social Distance Scale contains eight items and also uses a 5-point Likert response scale [44], and higher scores imply keeping less distance. The SDS has been reported with good internal consistency [47].

Behavior was measured by assessing participants’ confidence to help (5-point Likert scale), participants’ intention to help someone with mental illness (5-point Likert scale), and participants’ “first aid” actions helping someone with a psychiatric illness [48,49,50]. In order to measure confidence to provide help, a score was computed by summing up the answers as well as by calculating the proportion of two categories: “Likely confident”, if someone answered very confident, fairly confident, or neutral; and “Not confident”, if they answered = not confident at all or slightly confident. For participants’ first aid actions providing mental first aid to someone in needs, answers were “Yes” or “No”.

These outcomes, drawn from the original surveys conducted in Australia, were translated from their original English version to French, following Wild et al.’s process of well-principled translation [51].

Inter-rater reliability had significant to strong intra-class correlations for the six MHFA components (all *p* ≤ 001) and for the total MHFA score (r(49) = 0.81, *p* < 0.001) [40,52].

Participants’ mental health status was assessed by measuring their symptoms of quality of life with the Short-Form Health Survey (SF-12) [53]: perceived stress with the PSS-10 scale [54,55], depression with the PHQ-9 [56,57], generalized anxiety with the GAD-7 scale [58,59], and attention deficit hyperactivity disorder (ADHD) with the ADHD Self-Report Scale Screener (ASRS v1.1) [60,61]. Subjects were also asked a few questions related to their access to psychological services and any suicidal thoughts, plans, or attempts they had had. Participants had the option of selecting “I refuse to respond” for these questions.

Demographic questions included 14 items designed by the project leader and investigator that asked about subjects’ date of birth, gender, nationality, marital status, spoken languages, employment status (e.g., if they have a student/part-time job), housing situation, parental levels of education, and if they had already attended training on any mental health topics.

### 2.8. Statistical Analysis

Descriptive analysis included original means, standard deviations, and contingency tables to summarize the frequencies presented by three timepoints and two treatment groups (intervention vs. control).

We used Generalized Estimating Equations (GEE) [62,63] to examine the effects of groups (intervention vs. control) and timepoints (baseline, first follow-up and second follow-up) on knowledge, stigma, social distance, confidence to help, intention to help, and recognition of disorders. Performing MHFA action was excluded because it was only limited to those who had friends who had experienced mental health problems, and the sample size was too small.

GEE can be used to fit regression models to handle correlated outcomes when repeated measurements of the same subject are performed over time, and to adjust for missing values. The coefficients with standard errors for each timepoint, group, and time*group interaction are reported. GEE model-based multiple comparisons adjusting for the missing values were conducted. The mean differences were reported for continuous variables and the odds ratios were reported for categorical variables. The means and frequencies derived from the models varied slighted from the original means and frequencies after adjuring for missing values; the results are presented in Appendix A. The effect sizes for continuous variables were reported using the differences divided by the standard deviation of the model residuals, and the odds ratios were used for categorical variables.

For all analyses, the criterion for statistical significance was set as *p* ≤ 0.05, and all tests were two-tailed. We examined the potential bias of non-response by comparing the outcomes at baseline between those who participated at all three timepoints and those who failed to follow up, with no significant differences being found. We proceeded to complete case analyses under the assumption that the non-response mechanism was not related to the analyzed variable (missing completely at random MCAR). 

Data were analyzed using STATA 16.1 and R.4.03 (R Core Team. R: A language and environment for statistical computing (R Foundation for Statistical Computing. https://www.eea.europa.eu/data-and-maps/indicators/oxygen-consuming-substances-in-rivers/r-development-core-team-2006, https://cran.r-project.org/, accessed on 1 June 2022).

## 3. Results

### 3.1. Demographics

The participants’ mean age was 23.99 (*SD* = 4.17). Among the 107 participants, most were women (70%), and 57% worked part-time besides studying. Only seven participants reported having taken courses relating to mental health during their degree. The demographic characteristics did not differ significantly between the intervention and control groups (Table 1).

### 3.2. Comparing Knowledge, Attitude, and Behavior between the Control and Intervention Groups

The original means and frequencies over three timepoints by the two groups are presented in Table 2, and multiple comparisons are presented in Table 3.


**Situation at Baseline (T0)**


Results indicated non-significant differences between the control and intervention groups (I0-C0) in terms of recognition and knowledge about mental disorders, attitude and behavior, confidence to help, or first aid action before MHFA/ensa training (T0) (Table 3).

In both groups, less than 25% of the participants expressed confidence to help a person with a mental health disorder with similar mean scores at baseline (*M* = 2.70, *SD* = 0.86 vs. *M* = 2.81, *SD* = 0.89).


**Situation at 3 months after MHFA/ensa training (T1)**


Significant differences were apparent between the control and intervention groups 3 months after MHFA/ensa training (I1-C1), with improvements noted in knowledge mean scores, reduced stigma and social distance, and greater confidence to help in the intervention group. There were no statistically significant differences between groups in intention to help and recognition of a mental disorder even though the intervention group had higher mean scores or percentages than the control group. 


**Situation at 12 months after MHFA/ensa training (T2)**


Even 12 months after the training course, results indicated significant differences between the control group and the intervention group (I2-C2), with higher scores for knowledge, decreased stigma, and social distancing towards mental illness and more confidence to help in the intervention group. Similar to T1, no significant differences were found in intention to help and recognition of mental disorders).

In general, large effect sizes between the groups were found in knowledge and confidence and medium effect sizes were found in reduced stigma and social distance.

### 3.3. Assessing Changes over Three Time Points Using Generalized Estimating Equation (GEE) Control Group

The control group showed no significant differences (C1-C0, C2-C0) over time in all of the outcomes except intention to help, for which the participants reported higher mean scores at both T1 and T2.

#### Intervention Group

*Knowledge*. Relative to baseline (T0), participants in the intervention group reported higher means at both T1 and T2 in knowledge with medium effect sizes. Compared to T1, there was a slight reduction of the mean score at T2, however the difference was not statistically significant.

*Stigmatizing attitudes*. Participants showed substantial improvements in reduced stigma at T1 and T2 after the intervention (lower scores implied less stigma), both with large effect sizes. The mean scores increased slightly from T1 to T2, but the difference was not significant.

*Attitude-Social distance*: In terms of social distance towards a person with a mental illness, participants once again demonstrated significant improvements after the MHFA intervention (higher scores imply less social distance toward mentally ill person) at both T1 and T2 compared to T0, with medium effect size) and the difference between T1 and T2 was not significant.

*Behavior—Confidence to help*: After the intervention, greater mean scores were reported at both T1 and T2 with very large intervention effects (the effect size being larger than 1). We further dichotomized the “confidence to help” (feeling confident vs. likely/neutral/not confident) and found that a greater proportion of participants reported feeling confident to help someone with a mental disorder after the intervention, with large effect sizes at both T1 and T2 (odds ratio equal to 8.79 and 6.97, respectively).

*Behavior—Intention to help*: Participants reported higher mean scores on their intention to help at T1 compared to T0 with a large effect size (*d* = 0.91). Moreover, although the strength of the intervention’s effect reduced significantly from T1 to T2 (d = −0.53), a statistically significant medium effect remained 12 months after the intervention (*d* = 0.39).

*Recognition of disorder*: The proportion of participants who could recognize the disorder did not change after the intervention.

Table 4 shows the results of regression models using GEE to assess the intervention effects of timepoints, groups, and time*group interactions. For knowledge, stigma, social distance, and confidence to help, we found a significant interaction effect at T1, consisting in the intervention group having significantly improved results compared to the control group, with the significant interaction effect at T2 (12 months later) between the groups remaining only in reduced stigma, social distance, and confidence to help.

## 4. Discussion

This study is the first RCT in Switzerland to evaluate the impact of an MHFA/ensa training course on undergraduate students attending two different universities. Face-to-face MHFA/ensa program training was found to be both effective, sustainable, and feasible among Swiss undergraduate students.

The study revealed the significant short-term and long-term effectiveness of the MHFA training program. We found significant time*group interaction effects for the intervention group, increased basic knowledge about mental health and their confidence to help someone experiencing a mental health problem while decreasing the stigma towards mental illness compared to the control group. Interestingly, for reducing social distance toward those with a mental disorder, we found a marginal interaction effect at T1 but a very strong interaction effect at T2, with the difference between intervention and control groups being more pronounced at T2. For ‘intention to help’, we did not find significant time*group interaction effects but significant time effects, with both the intervention group and the control group increasing their scores of intention to help someone with a mental disorder at both T1 and T2. This can be partially explained by the study sample, with 74% of the students studying at the School of Nursing; the control group was also motivated to help patients without the MHFA course. Unfortunately, the sample size from the other school was too small to examine the difference between schools.

Although most of these effects decreased slightly over the time period between 3 months and 12 months after the intervention, these decreases were statistically non-significant and medium-to-large effect sizes were evident for most improvements at both T1 and T2.

These findings are consistent with those of other studies, especially regarding the significant increase in knowledge mean scores after the 12-h training [41,64]. Indeed, knowledge scores were increased 3 months after the intervention, and this increase was sustained through 12 months. Two RCT and two quasi-experimental studies, which measured the impact of an MHFA/ensa course (online or face-to-face) among undergraduate medical, nursing, and social work students, had already shown significantly increased mental health and mental health first aid knowledge among those who completed the intervention [26,41,64,65]. Two of these studies also found significant improvements in knowledge scores over time in the intervention group [26,65]. However, while the current study reported sustained improvement in knowledge 12 months after the intervention was completed, none of these previous studies assessed subjects beyond six months after the baseline. On the other hand, other studies conducted on the general public had documented sustained increases in knowledge from 6-month follow-up to a 2-year follow-up with medium effect sizes [24,27,28]. This said, further studies highlighted some attenuation in knowledge scores over time [41,66], suggesting that refresher courses might be of value.

In the present study, recognition of schizophrenia was greater in the control group versus the intervention group, both at baseline (T0) and at follow-up (T2). However, the proportion of participants who recognized schizophrenia correctly did not significantly differ between control and intervention groups at any of the three data-collection times (T0, T1, T2). In the intervention group, the proportion of students correctly recognizing schizophrenia remained at 47.2% after the intervention and only increased to 48.5% at the 12-month follow-up. Relative to other studies among nursing students [26,41], the proportion of our subjects who recognized the disorder correctly was low. This could be partially explained by roughly 30% of our students majoring in engineering or management, who therefore might have had less intrinsic awareness or knowledge about health subjects. Our results in schizophrenia recognition at baseline were closer to those reported for the general public in multiple studies worldwide [27,28,29].

The results of this study demonstrated a significant positive impact of MHFA/ensa training in decreasing personal stigma and social distance of participants towards persons with mental health disorders. These findings are consistent with those of other studies that demonstrated improvements over time [31,41,65]. Furthermore, both intervention and control groups showed low levels of stigmatizing attitudes at baseline. This may be explained by 49.5% of participants already being exposed to someone suffering from a mental illness and by awareness that has resulted from recent public awareness campaigns and social media [67]. One Brazilian cross-sectional study highlighted variables associated with the stigma towards psychiatric disorders, such as male sex, lower household income, a psychiatric disorder in a first-degree relative, internalized stigma and, specifically, fewer years of education [68]. Other studies have demonstrated associations between increased contact and/or information/knowledge about a mental health condition and someone’s willingness to care for and empathize with people with mental illness [69,70]. Results are also consistent with previously published findings regarding intention to help and first-aid actions. However, even if the score for intention to help and the percentage of participants who did help someone suffering from a psychiatric disorder increased from T0 to T1, the difference between the control and intervention groups was only significant for first aid action at T1.

Our study had limitations. First, females were highly overrepresented among the nursing students (86%). This, however, reflects the current demographic reality of the nursing workforce. We tried to include more males by enlisting subjects from another university to minimize gender bias. Unfortunately, the participation rate was low, mainly because many students already felt over-extended due to their class schedule and were unable to find time to participate. A few dropouts also occurred in the intervention group after the baseline assessment because of their inability to attend the 12-h course. However, few dropped out between T1 and T2, even though the last data collection happened during the COVID-19 shutdown.

This study also lacked a post-intervention assessment right after the course (less than 3 months after the baseline), but it did collect data 12 months afterwards to measure the long-term effects of the training, which prior studies had generally not done. 

For future research, embedding the MHFA/ensa course into students’ schedule as an optional course, or during semester breaks, could overcome the low participation rate we observed. The study population could also be extended to other universities and to first- year students to avoid potential biases in knowledge that might occur from prior courses within nursing, medical, social work, and other healthcare programs, wherein students would be expected to become more aware of mental health subjects over the course of their studies.

## 5. Conclusions

This RCT supports MHFA program training as both effective and sustainable among Swiss undergraduate students, even for students studying in health-related disciplines. Not only were improvements in several outcomes made three months after training, but these improvements were generally sustained, or only decreased slightly, after one year, showing that these effects can endure. The MHFA program was initially created for the general public but could be relevant and beneficial for young adults and students, especially for first-year undergraduate students. 

## Figures and Tables

**Figure 1 ijerph-20-01303-f001:**
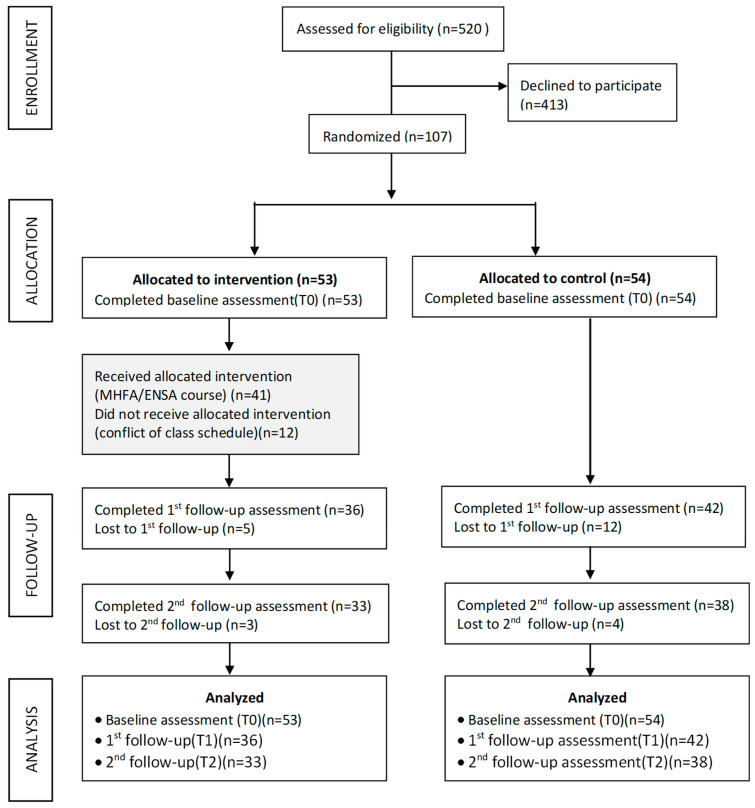
Flow diagram of participant recruitment.

**Table 1 ijerph-20-01303-t001:** Participant characteristics at baseline (T0).

Demographic Characteristics	Total (*N* = 107) *n* (%)	Control *n* (%)	Intervention *n* (%)
University			
HEIG-VD	33 (30.8)	17 (31.5)	16 (30.2)
La Source	74 (69.2)	37 (68.5)	37 (69.8)
Gender			
Women	75 (70.1)	38 (70.4)	37 (69.8)
Men	32 (29.9)	16 (29.6)	16 (30.2)
Student work			
Yes	61 (57.0)	31 (57.4)	30 (56.6)
No	46 (43.0)	23 (42.6)	23 (43.4)
Rate of work (n = 60)			
0–20%	40 (66.7)	23 (74.2)	17 (58.6)
>20%	20 (33.3)	8 (25.8)	12 (41.4)
Having taken Mental health course			
Yes	7 (6.5)	5 (9.3)	2 (3.8)
No	100 (93.5)	49 (90.7)	51 (96.2)
	**Total (*N* = 107)** **mean (*SD*)**	**Control** **mean (*SD*)**	**Intervention** **mean (*SD*)**
Age	23.99 (4.17)	24.22 (4.57)	23.75 (3.70)

**Table 2 ijerph-20-01303-t002:** Comparing knowledge, attitude, and behavior in Control and Intervention groups at three-time points.

Variables	T0 (*N =* 107)		T1 (*N =* 78)		T2 (*N =* 71)	
	Control mean (*SD*)	Intervention mean (*SD*)	Control mean (*SD*)	Intervention mean (*SD*)	Control mean (*SD*)	Intervention mean (*SD*)
MHFA knowledge	12.69 (2.00)	13.13 (1.79)	12.90 (1.91)	14.33 (1.59)	12.71 (2.01)	14.12 (1.88)
Attitude: Personal stigma	16.11 (2.95)	16.23 (3.07)	15.90 (2.70)	13.61 (2.71)	16.13 (3.05)	13.88 (2.94)
Social distance	24.00 (5.57)	25.51 (4.23)	24.83 (5.63)	28.72 (5.60)	24.32 (6.31)	28.82 (5.25)
Behavior: Confidence to help	2.70 (0.86)	2.81 (0.88)	2.88 (0.89)	3.83 (0.56)	2.94 (0.89)	3.76 (0.66)
Intention to help	2.16 (1.09)	2.17(1.09)	2.78 (1.61)	3.42 (2.03)	2.74 (1.24)	2.70 (1.13)
	**T0**		**T1**		**T2**	
	**Control** ***n* (%)**	**Intervention** ***n* (%)**	**Control** ***n* (%)**	**Intervention** ***n* (%)**	**Control** ***n* (%)**	**Intervention** **n (%)**
Recognition of disorder						
Yes	30 (55.6)	25 (47.2)	16 (38.1)	17 (47.2)	20 (52.6)	16 (48.5)
No	24 (44.4)	28 (52.8)	26 (61.9)	19 (52.8)	18 (47.4)	17 (51.5)
Confidence to help						
Yes	12 (22.2)	13 (24.5)	13 (31.0)	27 (75.0)	25 (65.8)	32 (97.0)
No	42 (77.8)	40 (75.5)	29 (69.1)	9 (25.0)	13 (43.2)	1 (3.0)
First aid action						
Yes	23 (82.1)	21 (84.0)	14 (63.6)	17 (100)	10 (83.3)	15 (93.8)
No	5 (17.9)	4 (16.0)	8 (36.4)	0	2 (16.7)	1 (6.3)

**Table 3 ijerph-20-01303-t003:** Multiple comparisons using Generalized Equation Estimation Model.

	Knowledge			Stigma			Social Distance			Confidence to Help			Intention to Help (Score)			Confidence to Help (Dichotomize)		Recognition of Disorder (Dichotomize)	
	Diff (SE)	*p*-Value	Effect Size	Diff(SE)	*p*-Value	Effect Size	Diff (SE)	*p*-Value	Effect Size	Diff (SE)	*p*-Value	Effect Size	Diff(SE)	*p*-Value	Effect Size	Odds Ratio [95%CI]	*p*-Value	Odds Ratio [95%CI]	*p*-Value
I0-C0	0.45 (0.36)	0.217	0.241	0.12 (0.58)	0.842	0.04	1.51 (0.95)	0.111	0.281	0.11 (0.17)	0.518	0.134	0.003 (0.21)	0.988	0.002	1.14 [0.46–2.49]	0.778	0.71 [0.33–1.53]	0.386
I1-C1	1.44 (0.39)	<0.001	0.774	−2.00 (0.59)	<0.001	−0.689	3.52 (1.21)	0.004	0.656	0.92 (0.16)	<0.001	1.139	0.63 (0.42)	0.128	0.462	6.52 [2.44–12.98]	<0.001	1.24 [0.51–3.06]	0.636
I2-C2	1.31 (0.44)	0.003	0.707	−1.94 (0.66)	0.003	−0.672	4.02 (1.22)	<0.001	0.75	0.73 (0.17)	<0.001	0.911	−0.042 (0.28)	0.881	−0.031	3.25 [1.13–6.02]	0.029	0.67 [0.27–1.69]	0.399
I1-I0	1.20 (0.34)	<0.001	0.647	−2.35 (0.39)	<0.001	−0.812	2.75 (0.96)	0.004	0.512	0.95 (0.13)	<0.001	1.179	1.24 (0.37)	<0.001	0.912	8.79 [3.40–18.16]	<0.001	0.90 [0.41–1.97]	0.784
I2-I0	0.95 (0.30)	0.002	0.512	−2.07 (0.44)	<0.001	−0.717	2.87 (0.77)	<0.001	0.535	0.86 (0.12)	<0.001	1.068	0.53 (0.23)	0.023	0.386	6.97 [3.16–17.26]	<0.001	0.88 [0.44–1.77]	0.729
I2-I1	−0.25 (0.36)	0.487	−0.135	0.28 (0.33)	0.407	0.096	0.12 (0.96)	0.899	0.023	−0.09 (0.10)	0.391	0.111	−0.72 (0.37)	0.05	−0.526	0.79 [0.35–1.92]	0.576	0.99 [0.61–1.60]	0.959
C1-C0	0.21 (0.34)	0.537	0.114	−0.24 (0.41)	0.555	−0.083	0.74 (0.72)	0.305	0.138	0.14 (0.14)	0.301	0.174	0.62 (0.30)	0.037	0.452	1.53 [0.72–3.96]	0.27	0.51 [0.25–1.05]	0.069
C2-C0	0.09 (0.35)	0.805	0.046	−0.02 (0.48)	0.975	−0.005	0.36 (0.75)	0.633	0.067	0.23 (0.14)	0.086	0.29	0.57 (0.25)	0.02	0.419	2.44 [0.92–4.89]	0.074	0.94 [0.44–2.00]	0.866
C2-C1	−0.12 (0.35)	0.733	−0.068	0.22 (0.34)	0.509	0.077	−0.38 (0.58)	0.512	−0.071	0.09 (0.12)	0.446	0.117	−0.05 (0.31)	0.885	0.033	1.59 [0.63–3.34]	0.329	1.81 [0.83–4.00]	0.135

**Table 4 ijerph-20-01303-t004:** Generalized Estimating Equations (GEE) to assess the effects of time, group and time*group.

	Knowledge		Stigma		Social Distance		Confidence to Help		Intention to Help		Recognition of Disorder	
	Estimate (SE)	*p*-Value	Estimate (SE)	*p*-Value	Estimate (SE)	*p*-Value	Estimate (SE)	*p*-Value	Estimate (SE)	*p*-Value	Estimate (SE)	*p*-Value
Intercept	12.69 (0.27)	<0.001	16.11 (0.40)	<0.001	24.00 (0.75)	<0.001	2.70 (0.12)	<0.001	2.17 (0.15)	<0.001	0.22 (0.27)	0.415
Time 1	0.21 (0.34)	0.537	−0.24 (0.41)	0.555	−0.74 (0.72)	0.305	0.14 (0.14)	0.301	0.62 (0.30)	0.037	−0.66 (0.37)	0.069
Time 2	0.09 (0.35)	0.805	−0.02 (0.48)	0.975	0.36 (0.75)	0.633	0.23 (0.14)	0.086	0.57 (0.25)	0.020	−0.07 (0.39)	0.866
Group	0.45 (0.36)	0.217	0.12 (0.58)	0.841	1.51 (0.95)	0.111	0.11 (0.17)	0.518	0.003 (0.21)	0.988	−0.34 (0.39)	0.386
Time1*Grp	0.99 (0.48)	0.041	−2.11 (0.56)	<0.001	2.01 (1.20)	0.095	0.81 (0.18)	<0.001	0.63 (0.47)	0.182	0.55 (0.54)	0.309
Time2*Grp	0.86 (0.46)	0.061	−2.06 (0.65)	<0.001	2.51 (1.07)	0.019	0.63 (0.18)	<0.001	−0.04 (0.34)	0.894	−0.06 (0.52)	0.912

Time 1: 1st follow-up; Time 2: 2nd follow-up; Group: 1 = intervention/0 = control; Time1*Grp: intervention group at 1st follow-up; Time2*Grp: intervention group at 2nd follow-up.

## Data Availability

The datasets analyzed in the current study are not publicly available due to the conditions specified in the data protection contract for this study.

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
