# Peer review of "The Effectiveness of Mental Health First Aid Training among Undergraduate Students in Switzerland: A Randomized Control Trial"

_ijerph, 2023, doi:10.3390/ijerph20021303_

Round 1

Reviewer 1 Report (Previous Reviewer 2)

The changes are appropriate and the results defensible.

This manuscript is a resubmission of an earlier submission. The following is a list of the peer review reports and author responses from that submission.

Round 1

Reviewer 1 Report

  1. The authors should carefully reread their manuscript as there are several spelling mistakes and ungrammatical sentences (including but not limited to: p.3 l. 143, p.6l.224, p.7l254, p.7l.267,p. 8 l297-298,p.9 l346-348)
  2. I am a little confused about the time line of the study.The authors should provide more detail regarding the time points of assessments for both study groups. E.g. the authors state that all participants completed a baseline survey (T0) but do not mention at what time point this survey was completed? Was it the same time point for all participants (intervention+control)? The authors further state, that the intervention took place some time between Oct.28 and end of Dec. and that the intervention group completed the follow up survey (T1) 3 month after the intervention was completed. But what about the control group?
  3. P. 3l.140 “After the Baseline survey, both groups received emails with a link to the questionnaires” it is unclear what questionnaires the authors are referring to.
  4. One of the aims of the study is to “examine its short-term and long-term effects […] on […] mental health status […]” p. 2-3 l.97-99. However. After the description of the outcome measures the participants’ mental health status is not mentioned again. Neither in the results nor in the discussion. Thus, the authors should either exclude this variable from their aims/introduction and measures or include the results and discuss them accordingly.
  5. If the authors choose to report the results of participants’ mental health status they need to take into account:
  • that the students were likely experiencing very different circumstances during the different time points of the assessments (e.g. exam period or semester break, the pandemic…)
  • that they did not compare the MHF intervention to another type of intervention(the control group did nothing), thus potential positive effects may be simply due to participants’ expectation that their participation increases their mental health.
  1. All effect sizes should be reported in the results.
  2. The authors conclude that the MHFA program could be even more relevant and beneficial for young adults. However as they did not compare the effect of the MHFA program between young adults and adults, I feel that this conclusion is not appropriate.
  3. The authors state at multiple occasions, that the MHFA effectively increases the knowledge about mental illness. However, the questionnaire used is “based on the content of the MHFA/ensa course” (p.5 l. 179) rather than including a questionnaire that generally assesses mental health. It is not really surprising that answers to questions that are specifically based on the content of a course improve after the course was visited. Supporting this is the fact that the only general assessment (the identification of a mental illness) does not improve. Thus, I believe that the authors should be more careful in their statement that the program effectively increases knowledge about mental illness.  

Reviewer 2 Report

This paper details the efficacy of a mental health training programme taken by undergraduates in Switzerland. It is generally well written, clearly informed by relevant literature, and could make something of a contribution to the literature on MHFA.   However, while the overarching (RCT) design is well delivered, the analytical strategy is questionable. I would expect the continuous data to be analysed as a series of two factor mixed anova tests, with some post-hoc analysis - the between group is the treatment (x2), and the within group is time (x3). The advantage of using such an approach is that it will be able to determine the interaction between treatment and time - which is essentially what the paper aims to do. Also, because a two factor mixed anova is a groupwise comparison test it greatly reduces the chances of a type 1 error, which is highly likely here given the number of t-tests. The anova test is also relatively robust even where the data might violate some of the standard assumptions.   More generally, the results of the analysis are also not reported in APA style, and this makes them difficult to interpret (particularly the organisation of the table 2, which repeats information), and I'm also not sure why the descriptive statistics have been subjected to inferential testing.   The analysis of the binary data is probably OK (I'm presuming a series of three McNemar tests), but could be presented more clearly (T0/T1; T0/T2; T1/T2). Other analytical options are available here, but they are increasingly complicated and unlikely to add to the analysis.     I don't actually think that making these changes would take too much time, but it would improve the clarity of the analysis. 

Reviewer 3 Report

This study tested the short-term and long-term effectiveness of a mental health awareness training program (Mental Health First Aid, MHFA) by a randomized control trial in Swiss undergraduates. Results showed that MHFA training did improve mental health knowledge, reduce stigma and increase behavior intentions among intervention groups, and the effect lasted after 12-months. The study is generally well-done; manuscript well-written. I have some improvement suggestions:

  • In abstract, “… mainly due to lack of knowledge, perceived need, and stigmatization toward mental illness” [Line 13]. The treatment gap is due to lack of perceived need, or due to perceived need?
  • The introduction could be tightened up a little, especially the first three paragraphs. The severity of mental disorders among youths, the existence of treatment gap and the causes of such gap are all background information. The focus of the introduction should be on the MHFA.
  • The mental health conditions of participants were measured, but seem missing in the analysis. Is their mental health condition in any way affecting the participation (vs. attrition) in the MHFA, or affecting the effectiveness of MHFA?
  • The differences between Intervention and Control groups at T0, T1 and T2 could be better analyzed by mixed-design analyses of variance (for continuous variables) or generalized linear mixed models (for categorical variables).
  • The asterisks (*) in Table 2 and Table 3 are confusing. It is not clear whether they signify differences between intervention and control at the same time point or across time points. Authors did note that “Significant difference between Control and Intervention groups” at Table 2, but if so, labeling asterisks on both groups still remains visually confusing. Usually authors may add another column with p values and label asterisks there; or if the space is limited, try boldface on both values rather than two asterisks.
  • Adding several graphs would be clearer at illustrating the short- and long-term effectiveness of the MHFA.

Reviewer 4 Report

Dear authors,
First of all, congratulations on your work. There is a pressing need to raise awareness of mental health problems, eliminate stigma and promote rapid access to health systems.
I loved the work, however, I would like to make some comments:
1- In the method you indicate that you have performed with the experimental group however you have not indicated that you have performed the control group. I imagine that they have been on standby without performing any activity. Please indicate it clearly in the text for the reader to have the certainty. 
2- You have used several instruments to measure all the variables of the study, but I think it is appropriate to indicate the validity and reliability statistics of these instruments; either in the original samples of the authors or those found in your sample.
3- Due to the progressive loss of participants (normal in this type of studies) in the study, it would be advisable to make calculations on the statistical potential to know if there is enough sample to carry out the analyses.
4- The authors have opted to perform multiple Student's t-tests; however, I consider that a repeated measures mixed ANOVA design (2x3) would provide greater strength to the results.
 5- In line 315 it is stated that studies with similar results have been found, however no study is cited. 

kind egard

Round 2

Reviewer 3 Report

From the ANOVA results, it is apparent that the time*group interaction is not significant. Reporting only pairwise t tests in the manuscript is tantamount to cherry-picking methods that produce significant results to suit their conclusion, which is unacceptable. Not mention the inflated Type I error rates in repeatedly carrying out t tests. I suggest authors should replace the series of t tests (the table became more cumbersome with three columns of means and three columns of t test results now; See Table 3) with ANOVA or mixed linear model.

As for their argument that they are interested in short-term vs long-term effects, the statistical issue remains. From the ANOVA table they provided to Reviewer 2’s point 1, there is indeed an increase in DV in Time 2 vs Time 1; and that increase leveled off (i.e., not significant; rather than a decrease, as they claimed in Response). This could be understood as a case of quadratic trend in the Time variable; if that is their intention, they should use polynomial component in conducting their mixed-design ANOVA.

As for the participant loss, it is then advisable to conduct mixed linear model, which can handle missing cases well; the core hypothesis could be answered by examining the cross-level interaction.  

Reviewer 4 Report

Dear Authors 
thank you for taking into account the suggestions made
Kind regard